# A Lipid Nanoparticle-Formulated Self-Amplifying RNA Rift Valley Fever Vaccine Induces a Robust Humoral Immune Response in Mice

**DOI:** 10.3390/vaccines12101088

**Published:** 2024-09-24

**Authors:** Paul K. Kitandwe, Paul Rogers, Kai Hu, Owen Nayebare, Anna K. Blakney, Paul F. McKay, Pontiano Kaleebu, Robin J. Shattock

**Affiliations:** 1MRC/UVRI & LSHTM Uganda Research Unit, Plot 51-59 Nakiwogo Road, Entebbe P.O. Box 49, Uganda; owen.nayebare@mrcuganda.org (O.N.); pontiano.kaleebu@mrcuganda.org (P.K.); 2Department of Infectious Diseases, Imperial College London, Norfolk Place, London W2 1PG, UK; paul.rogers@imperial.ac.uk (P.R.); kai.hu@chimeris.uk (K.H.); anna.blakney@msl.ubc.ca (A.K.B.); p.mckay@imperial.ac.uk (P.F.M.); r.shattock@imperial.ac.uk (R.J.S.); 3Uganda Virus Research Institute, Plot 51-59, Nakiwogo Road, Entebbe P.O. Box 49, Uganda

**Keywords:** self-amplifying RNA, saRNA, RVF vaccine, VEEV, Rift Valley fever virus, RVFV, lipid nanoparticles

## Abstract

Rift Valley fever (RVF) is a mosquito-borne viral zoonosis that causes high fetal and neonatal mortality rates in ruminants and sometimes severe to fatal complications like encephalitis and hemorrhagic fever in humans. There is no licensed RVF vaccine for human use while approved livestock vaccines have suboptimal safety or efficacy. We designed self-amplifying RNA (saRNA) RVF vaccines and assessed their humoral immunogenicity in mice. Plasmid DNA encoding the Rift Valley fever virus (RVFV) medium (M) segment consensus sequence (WT consensus) and its derivatives mutated to enhance cell membrane expression of the viral surface glycoproteins n (Gn) and c (Gc) were assessed for in vitro expression. The WT consensus and best-expressing derivative (furin-T2A) were cloned into a Venezuelan equine encephalitis virus (VEEV) plasmid DNA replicon and in vitro transcribed into saRNA. The saRNA was formulated in lipid nanoparticles and its humoral immunogenicity in BALB/c mice was assessed. High quantities of dose-dependent RVFV Gn IgG antibodies were detected in the serum of all mice immunized with either WT consensus or furin-T2A saRNA RVF vaccines. Significant RVFV pseudovirus-neutralizing activity was induced in mice immunized with 1 µg or 10 µg of the WT consensus saRNA vaccine. The WT consensus saRNA RVF vaccine warrants further development.

## 1. Introduction

Rift Valley fever (RVF) is a severe viral zoonosis that causes high rates of fetal and neonatal deaths in ruminants and fatal hemorrhagic fever or other serious complications in some infected humans [1]. This disease, which is caused by the Rift Valley fever virus (RVFV), is transmitted by mosquitoes and in humans, and also through contact with the blood, body fluids, or tissues of infected animals [2]. RVF is endemic to Africa and some countries in the Arabian Peninsula like Saudi Arabia and Yemen [3]. The possibility of RVF spreading to new geographical regions exists due to the presence of several risk factors that favor its spread in those areas such as the competent mosquito vectors, the wide range of susceptible domestic and wild animals, and climate change [4].

Despite the serious threat that RVF poses to public health and the livestock industry, it has no licensed vaccine for human use or one that is universally acceptable for veterinary use. Licensed livestock RVF vaccines are all either live-attenuated or inactivated, but the former are associated with teratogenicity and residual virulence while the latter have poor immunogenicity [5]. Additionally, nearly all licensed RVF vaccines lack the differentiating infected from vaccinated animals (DIVA) property, making their use in RVF non-endemic countries unlikely as it would complicate RVFV surveillance and control and hinder the export of animals vaccinated with such vaccines [6].

Due to the limitations of the currently licensed RVF vaccines, substantial research is ongoing to develop safer and more efficacious vaccines against this zoonosis using various vaccine platforms such as genetically modified live-attenuated virus, DNA, protein subunits, viral vectors, viral replicons, and virus-like particles [5]. Given that RVF outbreaks are sporadic and primarily occur in low-income countries, it is highly desirable to develop vaccines against this zoonosis using platforms that enable low-cost, simple, and rapid production. As demonstrated with COVID-19, messenger RNA (mRNA) is a suitable platform for the rapid development of vaccines against emerging infectious diseases. This is due to its rapid production, flexibility that allows for the swift adaptation to new viral variants, scalability to meet increasing demand and a good safety and immunogenicity profile [7,8].

Self-amplifying RNA vaccines represent a next-generation mRNA vaccine platform that utilizes the genome of alphaviruses such as Sindbis virus (SINV), Semliki Forest virus (SFV), or Venezuelan equine encephalitis virus (VEEV) to replicate itself. In this vaccine format, the sequence encoding the alphaviral structural sequence under the control of the 26S subgenomic promoter is replaced with a heterologous gene of interest (GOI). Upon delivery into a host cell, the GOI is amplified by the RNA-dependent RNA polymerase (RdRp) complex synthesized from the viral genome’s 5′ encoded nonstructural proteins nsP1-4 [9,10]. Compared to conventional mRNA vaccines, saRNA requires a significantly lower dose to achieve similar antigen expression levels [11]. This can reduce the vaccine production costs and side effects [12]. Furthermore, saRNA vaccines have a much longer antigen expression duration than conventional mRNA vaccines, which may enhance the overall immune response [13].

The precise mechanistic correlates of immune protection against RVFV infection are not fully understood. However, neutralizing antibodies targeting the RVFV surface glycoproteins n (Gn) and c (Gc) can effectively prevent viral infection, making them suitable vaccine antigens [14,15,16]. These glycoproteins, which are encoded by the RVFV M segment, are not efficiently delivered to the cell surface due to a Golgi apparatus-localizing signal and endoplasmic reticulum retention motif located in the cytoplasmic tails of Gn and Gc, respectively [17]. Removal or mutation of these cytoplasmic tail sequences may enhance the cell membrane expression of Gn and Gc [18,19]. This study designed and evaluated the humoral immune responses of BALB/c mice to two lipid nanoparticle (LNP)-formulated VEEV genome-based saRNA candidate vaccines encoding RVFV Gn and Gc. The vaccine encoding the consensus surface glycoprotein sequence induced significant RVFV pseudovirus-neutralizing activity unlike the one with mutations that enhanced the in vitro cell membrane expression of these glycoproteins.

## 2. Materials and Methods

### 2.1. Generation of the Consensus RVFV M Segment Sequence

The coding sequence of all complete and near-complete RVFV M segment sequences deposited in GenBank and the Virus Pathogen Database and Analysis Resource (ViPR) as of 31 December 2018 were aligned in SnapGene software version 4.2.0 (Dotmatics, Boston, MA, USA) using the MUSCLE algorithm. The aligned sequences were edited to remove gaps, and the long ones were trimmed to cover the 3594 bp coding region producing a consensus RVFV M segment coding sequence at a 50% threshold. This sequence was translated and truncated to start from the fourth methionine, resulting in a consensus RVFV surface glycoprotein amino acid sequence.

#### Phylogenetic Analysis of the RVFV Wild-Type Consensus Sequence

Phylogenetic analysis was conducted to compare the RVFV M segment consensus sequence with the sequences used to generate it. The truncated RVFV M segment coding sequences were aligned with the consensus sequence using SnapGene (Appendix A)and exported to MEGA X software version 10.0.5. In MEGA X, a phylogenetic tree was generated using the maximum likelihood algorithm using the Tamura-3 parameter substitution model with gamma distribution [20]. Tree branch reliability was estimated using bootstrap analysis with 1000 replicates. Classification of the RVFV strains followed that used by Samy, Peterson, and Hall, 2017 [3].

### 2.2. Generation of Mutated RVFV M Segment Sequences

Various mutations were made to the RVFV M segment consensus sequence to increase the plasma membrane expression of Gn and Gc or to alter their conformation as a strategy to increase the immunogenicity of these glycoproteins (Figure 1).

Various mutations were made to the Rift Valley fever virus (RVFV) medium (M) segment: (1) WT consensus, an RVFV M segment consensus. (2) K1064A, a construct with lysine-to-alanine substitution at position 1064 shown to cause a mis-localization of RVFV surface glycoprotein n (Gn) to the plasma membrane [21]. (3) H727A, a construct having a histidine-to-alanine substitution at position 727 designed to stabilize RVFV surface Gn and c (Gc) heterodimer in its pre-fusion conformation [22]. (4) L202C_F672C (Gn-S-Gc), a construct with leucine-to-cysteine and phenylalanine-to-cysteine substitutions at positions 202 and 672, respectively, designed to introduce disulfide bonds between Gn and Gc. (5) L202C_F672C_H727A (H727A-S), a construct combining mutations for constructs 3 and 4. (6) K1050del, a construct lacking the Gc cytoplasmic tail shown to cause translocation of some Gn and Gc to the plasma membrane. (7) Furin-T2A, a construct with cytoplasmic tail deletions in both the Gn and Gc and with a furin enzyme cleavage and a T2A self-cleaving peptide separating these glycoproteins. This construct, which lacks the Golgi localization signal of Gn and the endoplasmic reticulum (ER) targeting signal of Gc, was designed to increase the plasma membrane expression of Gn and Gc and induce their efficient cleavage. (8) Gn, a construct encoding Gn only with its native signal sequence replaced with an artificial signal peptide MDRAKL_10_PQAQA, designed to increase extracellular expression of Gn. All constructs were truncated to start from the fourth methionine of the complete RVFV M segment sequence, which results in the translation of Gn and Gc only [21].

To make these mutations, the RVFV M segment consensus sequence was cloned into the mammalian expression vector pcDNA™3.1 (+) using GeneArt gene synthesis (Thermo Fisher Scientific, Cambridge, UK), and its maxiprep prepared using the Qiagen plasmid plus maxi kits (Qiagen, Hilden, Germany). The desired mutations were introduced into the RVFV M segment consensus sequence using Invitrogen GeneArt Strings DNA Fragments (Invitrogen, Waltham, MA, USA). These custom-made double-stranded linear DNA fragments were cloned into the pcDNA3.1+ vector carrying the consensus RVFV M segment sequence using the NEBuilder HiFi DNA Assembly kit (New England Biolabs, Ipswich, MA, USA), producing plasmids with the desired RVFV M segment mutations.

### 2.3. Plasmid DNA Transfection

Human embryonic kidney (HEK) 293T/17 adherent cells (ATCC CRL-11268, Manassas, VA, USA) and FreeStyle 293-F suspension cells (Gibco, Waltham, MA, USA) were transfected with polyethyleneimine (PEI) and 293fectin (Invitrogen) reagents, respectively. For PEI transfection, a day before transfection, HEK 293T/17 cells suspended in complete Dulbecco’s Modified Eagle’s Medium (DMEM) [DMEM supplemented with 10% fetal bovine serum (Sigma, St. Louis, MO, USA), 2mM L-glutamine (Sigma), and 1% penicillin-streptomycin (Sigma)] were plated onto 6-well tissue culture plates at a density of 3.0 × 10^5^ cells per well and incubated overnight at 37 °C and 5% CO_2_. The different RVF plasmid DNA (pDNA) constructs were mixed with PEI at a DNA-to-PEI ratio of 1:3 and incubated for 10 min at room temperature. The mixture was slowly added to HEK293T/17 cells, which were then incubated at 37 °C and 5% CO_2_ for 48 h.

Transfection using 293fectin transfection reagent was carried out following the manufacturer’s instructions. Briefly, the viable cell counts of the HEK 293-F cells to be transfected were determined using trypan blue dye exclusion. RVFV pDNA was incubated with 293fectin for 30 min after dilution in Opti-MEM I reduced serum medium (Invitrogen). The pDNA-293fectin complex was then added to 1.0 × 10^6^ HEK 293-F cells in a 6-well plate, which was then topped up to 5 mL with FreeStyle 293 expression medium (Invitrogen). The cells were then incubated at 37 °C, 8% CO_2,_ and 125 rpm in a humidified incubator for 48 h.

### 2.4. Synthesis of RVFV Self-Amplifying RNA

A pDNA vector based on a Trinidad donkey VEEV alphavirus genome was used to synthesize saRNA replicons. Linear DNA fragments encoding the RVFV M segment consensus sequence and the furin-T2A sequences custom-synthesized by GeneArt Gene synthesis (Thermo Fisher Scientific, UK) were cloned into the VEEV pDNA vector using the NEBuilder HiFi DNA assembly kit following the manufacturer’s instructions and successful clones confirmed by Sanger sequencing. SaRNA constructs encoding RVFV surface glycoproteins (RVFV saRNA) were then synthesized by in vitro transcription of these pDNA clones. The pDNA vector was first linearized by MluI restriction enzyme digestion followed by in vitro transcription using the MEGAScript T7 transcription kit (Invitrogen). The transcripts were purified by lithium chloride precipitation and then capped using the ScriptCap Cap 1 capping system Kit (Cellscript Inc., Madison, WI, USA) following the manufacturer’s instructions. The capped transcripts were also purified by lithium chloride precipitation, re-suspended in RNA storage buffer (10 mM HEPES, 0.1 mM EDTA, and 100 mg/mL trehalose), and stored at −80 °C until further use. The concentration and purity of the resultant RVFV saRNA were measured on a NanoDrop One spectrophotometer (Thermo Fisher Scientific, Madison, WI, USA) and its integrity was assessed using a denaturing agarose gel.

### 2.5. Self-Amplifying RNA Transfection

Lipofectamine MessengerMAX (Thermo Fisher Scientific, Waltham, MA, USA) and RVFV saRNA were both diluted with Opti-MEM I and incubated together for five minutes. The mixture was then added to the HEK293T/17 cells plated a day earlier at a density of 3.0 × 10^5^ cells per well. The transfected cells were then incubated at 37 °C and 5% CO_2_ for 24 h.

### 2.6. Assessment of In Vitro Expression Using SDS-PAGE and Western Blot

HEK293T/17 cells transfected with RVFV pDNA or RVFV saRNA were lysed on ice with Pierce IP lysis Buffer (Thermo Scientific, Waltham, MA, USA) in the presence of a protease inhibitor cocktail (Abcam, Boston, MA, USA). For the reduced Western blot, samples were reduced by heating at 70 °C for 10 min after suspension in lithium dodecyl sulfate (LDS) sample buffer (Invitrogen) and sample reducing agent (Invitrogen). The sample-reducing agent contains dithiothreitol (DTT) that helps break the disulfide bonds to enable the proper separation of Gn and Gc by size. Non-reduced Western blot samples were neither heated nor treated with a sample-reducing agent. Sodium dodecyl sulfate (SDS) polyacrylamide gel electrophoresis-(PAGE) was performed on the cell lysate using the Bolt 4–12% Bis-Tris Plus gels (Invitrogen) which was run at 200 volts for 35 min in SDS running buffer (Invitrogen). The separated proteins were transferred onto a methanol-activated polyvinylidene fluoride (PVDF) membrane by electrotransfer at 35 volts for one hour in SDS transfer buffer (Invitrogen). For reduced samples, bolt antioxidant (Invitrogen) was added to the SDS running buffer and transfer buffer to prevent sample reoxidation. After sample transfer, the membrane was blocked for one hour at room temperature in 5% non-fat dried milk powder in phosphate-buffered saline (PBS) + 0.05% Tween 20. The membrane was incubated at 4 °C overnight with a rabbit anti-RVFV Gn Immunoglobulin G (IgG) monoclonal antibody RV-Gn1 [23], mouse anti-RVFV Gn IgG monoclonal antibody clones 3C10 or 4D4 (both BEI Resources, Manassas, VA, USA), or mouse anti-RVFV Gc monoclonal antibody clones 1G4 or 9C10 (both BEI Resources). The mouse anti-RVFV Gn and anti-RVFV Gc monoclonal antibodies were obtained from the Joel M. Dalrymple-Clarence J. Peters USAMRID antibody collection through BEI Resources, NIAID, NIH. The anti-beta actin mouse monoclonal antibody (Invitrogen) was used as the loading control. All antibodies were diluted to a concentration of 1 μg/mL in 5% non-fat dried milk powder and PBS + 0.05% Tween 20. After incubation, the membrane was washed three times for 5 min each with PBS + 0.05% Tween 20 and then incubated for one hour at room temperature with the secondary antibody: goat anti-rabbit IgG horseradish peroxidase (HRP) (Invitrogen) or goat anti-mouse IgG HRP (Invitrogen) diluted 1:10,000 in 5% non-fat dried milk powder in PBS+0.05% Tween 20. Another wash as described above was performed and the membrane was incubated with Immobilon Crescendo Western HRP substrate (Millipore, Burlington, MA, USA) for three minutes at room temperature. The membrane was then read using the biostep Celvin S chemiluminescence reader (Celvin S 420 FL, biostep, Burkhardtsdorf, Germany).

### 2.7. Assessment of In Vitro Expression of Gn and Gc Using Flow Cytometry

Transfected cells were counted using trypan blue dye exclusion and stained with live/dead fixable violet (Invitrogen) according to the manufacturer’s instructions. After 20 min of room temperature incubation, the cells were centrifuged in 1 mL PBS and stained with the same primary antibodies used for the Western blot diluted 1:100 in PBS. After another 20 min incubation on ice, the cells were washed in 1 mL PBS and incubated with a secondary antibody, goat anti-rabbit IgG PE (Santa Cruz Biotechnology, Dallas, TX, USA) or goat anti-mouse IgG PE (Abcam), diluted 1:100 in PBS. The cells were then incubated for 20 min on ice, washed as before, and re-suspended in 400 µL of PBS. The cells were then acquired on a BD LSR Fortessa flow cytometer (BD Biosciences, San Jose, CA, USA) and the expression of Gn and Gc was measured as median fluorescence intensity (MFI) using FlowJo™ v10.7 Software (BD Life Sciences, East Rutherford, NJ, USA).

### 2.8. Formulation of saRNA

The saRNA was formulated in LNP using a self-assembly process whereby an aqueous solution of saRNA was rapidly mixed with an ethanolic solution of the LNPs. The LNPs used and their method of formulation were similar to those described by McKay et al., 2020 [24]. They contained an ionizable cationic lipid (proprietary to Acuitas), phosphatidylcholine, cholesterol, and polyethylene glycol lipid. The proprietary cationic ionizable lipid and LNP composition are described in US patent US10,221,127.

### 2.9. Mice Immunization

A total of 35 six-week-old BALB/c mice were randomly assigned to seven groups of 5 mice each. After a one-week acclimation period, each mouse was immunized intramuscularly with 50 μL of either 0.1 μg, 1.0 μg, or 10 μg of the candidate saRNA RVFV vaccine encoding either the WT consensus or the Furin-T2A sequence. The mice in the negative control group received 10 μg of the saRNA RVFV vaccine encoding the rabies glycoprotein. After 28 days, the mice were administered a booster immunization using the same dose and route. Blood was collected from the mice on days 14, 28, and 35 to conduct RVFV Gn IgG ELISA and assess RVFV pseudovirus-neutralizing activity.

This study was conducted at Imperial College London St. Mary’s Campus after obtaining ethics approval from its Animal Welfare and Ethics Review Board. It was conducted following the UK Home Office Animals (Scientific Procedures) Act 1986 under the project license (PPL) number P63FE629C (P.F.M.).

### 2.10. RVFV Antigen-Specific IgG ELISA

RVFV surface antigen IgG titers in mouse sera were assessed by a semi-quantitative ELISA. In brief, ELISA plates were coated with 100 μL per well (1 μg/mL in PBS) of recombinant Gn protein from RVFV strain MP12 (Sino Biological, Beijing, China). For the standard, 100 μL per well (1 μg/mL in PBS) of goat anti-mouse IgG kappa (Southern Biotech, Birmingham, AL, USA) and lambda chains (Southern Biotech) were used. After overnight incubation at 4 °C, the plates were washed four times with PBS + 0.05% Tween 20 and blocked for one hour at 37 °C with 200 μL per well of assay buffer (PBS + 0.05% Tween 20 with 1% bovine serum albumin). After washing the plates as before, 50 μL of mice sera, diluted 1:100, 1: 1000, and 1: 10,000 in assay buffer, and a 5-fold dilution series of the IgG standard (Southern Biotech, Birmingham, AL, USA) were added per well in triplicate starting with a 1000 ng/mL dilution. The plates were then incubated for one hour at 37 °C, washed as before, and 100 μL of goat anti-mouse IgG human adsorbed-HRP (Southern Biotech) diluted 1:2000 in assay buffer was added per well. After incubation at 37 °C for one hour, the plates were washed as before, and 50 μL of KPL sure blue 3,3′,5,5′-Tetramethylbenzidine microwell peroxidase substrate (Sera care, Milford, MA, USA) was added per well. After 5 min, the reaction was stopped using 50 μL per well of 0.12 N HCl acid stop solution (Sera care), and the absorbance of each well was measured spectrophotometrically at 450 nm.

### 2.11. RVFV Pseudovirus Neutralization Assay

An HIV-pseudotyped luciferase-reporter-based system was used to assess the neutralization capacity of sera from vaccinated mice. The RVFV pseudoviruses were produced by co-transfection of 293T/17 cells with an HIV-1 gag-pol plasmid (pCMV-Δ8.91, a kind gift from Prof. Julian Ma, St George’s University of London), a firefly luciferase reporter plasmid (pCSFLW, a kind gift from Prof. Julian Ma, St George’s University of London), and a plasmid encoding the RVFV M segment consensus sequence at a ratio of 1:1.5:1. The virus-containing medium was clarified by centrifugation and filtered through a 0.45 μm membrane 72 h after transfection, and subsequently aliquoted and stored at −80 °C. For the neutralization assay, heat-inactivated sera were serially diluted and incubated with the RVFV pseudovirus for one hour at 37 °C. Then, 1.0 × 10^5^ HEK293T/17 cells were added to the serum–virus mixture and cultured at 37 °C in 5% CO_2_ for 48 h. The luciferase activity from the cells was then measured using the bright-glo luciferase assay system (Promega Corp., Madison, WI, USA), and the half-maximal inhibitory concentration (IC_50_) neutralization was calculated.

### 2.12. Statistical Analysis

Data were analyzed using GraphPad Prism version 10.1.0 for Mac, GraphPad Software, Boston, MA, USA, https://www.graphpad.com (accessed on 9 September 2024). Statistical differences were calculated using either a Kruskal–Wallis test adjusted for multiple comparisons or a two-way ANOVA adjusted for multiple comparisons. A *p*-value of <0.05 was considered significant.

## 3. Results

### 3.1. Phylogenetic Analysis of the RVFV M Segment Consensus Sequence

The phylogenetic analysis showed that the RVFV wild-type consensus sequence clustered with RVFV strains isolated from East Africa (Uganda, Kenya, Tanzania), Sudan, Madagascar, and Saudi Arabia (Figure 2). These isolates were from RVFV outbreaks that occurred at the turn of the century. The RVFV wild-type consensus and the live attenuated RVF vaccine strains all clustered in different clades. The RVFV wild-type consensus, MP-12, clone 13, and Smithburn vaccine strains clustered with strains in clades I, A, K, and E respectively. These results show that the wild-type consensus vaccine is more representative of the more recent circulating RVFV strains from outbreaks that occurred in the more recent past as opposed to the strains that have been used to make current live-attenuated vaccines from the 1940s (Smithburn) and 1970s (Clone 13 and MP12). Therefore, a vaccine based on a wild-type consensus sequence is more likely to induce antibodies that will effectively neutralize the RVFV strains most relevant to public health and the livestock industry today.

### 3.2. In Vitro Expression of RVFV pDNA Using SDS-PAGE and Western Blot

Following transfection of HEK293 cells with pDNA encoding RVFV Gn and Gc, SDS-PAGE Western blot assays were performed on the cell lysates to assess in vitro antigen expression. For the reduced denatured cell lysates, RVFV Gn expression was observed in all pDNA constructs (predicted size, 61 KDa) (Figure 3A). The size of Gn expressed from the Gn only and furin-T2A construct was smaller than that of the rest of the constructs. This conformed with their predicted sizes of 49 KDa and 51 KDa, as expected from their amino acid truncations (see Figure 1). For the non-reduced denatured cell lysates, strong Gn expression was observed for the WT consensus, K1064A, K1050del, and H727A constructs, while weak expression was observed for furin-T2A and very weak expression for the Gn-S-Gc and H727A-S constructs (Figure 3B). No Gc expression was observed from the reduced cell lysates of all constructs (predicted size, 56 kDa) (Figure 3C). However, when the samples were not reduced, Gc expression was observed for the WT consensus, K1064A, and K1050del constructs (Figure 3D). (Appendix A for raw data files).

### 3.3. In Vitro Cell Surface Expression of RVFV pDNA Using Flow Cytometry

Flow cytometry was used to assess the cell surface expression of RVFV Gn and Gc from HEK293 cells 48 h after transfection with the different RVFV pDNA constructs. All constructs except the cytoplasmic tail- and transmembrane domain-deficient Gn construct expressed RVFV Gn on the cell surface (Figure 4A). Significant RVFV Gn expression was observed for the furin-T2A (MFI = 1457, SD = 187, *p* = 0.004) and K1050del (MFI = 1030, SD = 454, *p* = 0.013) constructs. These two were the only ones with a higher MFI than the WT consensus construct. For RVFV Gc, increased expression was observed for K1050del, K1064A, Gn-S-G, H727A, and H727A-S, although significant expression was only shown by H272A (MFI = 2147, SD = 127, *p* = 0.04). Surprisingly, RVFV Gc expression from the WT consensus and the furin-T2A constructs was similar to that of the untransfected samples (Figure 4B).

### 3.4. In Vitro Expression of RVFV Gn from saRNA

After an assessment of the RVFV Gn and Gc expression of the different RVFV pDNA constructs, the furin-T2A constructs that had the highest cell surface expression of RVFV Gn and the WT consensus were synthesized into saRNA. Linear DNA fragments encoding the DNA sequences of these constructs made by Invitrogen’s GeneArt gene synthesis were assembled into a VEEV pDNA vector using Gibson assembly cloning. The cloned product was transcribed to saRNA using the MEGAScript T7 RNA polymerase transcription kit followed by capping using the ScriptCap Cap 1 capping system kit. Following transfection of HEK293 cells with this saRNA, an SDS-PAGE Western blot was performed on the whole-cell lysates under reducing conditions to assess in vitro antigen expression. Strong expression was seen with the wild-type consensus while furin-T2A showed very weak expression (Figure 5A). The cell surface expression of RVFV Gn from HEK293 was assessed using flow cytometry 24 h after transfection with the saRNA. The cells were stained with mouse or rabbit anti-RVFV Gn IgG monoclonal antibodies and mouse or rabbit IgG PE primary and secondary antibodies, respectively. In contrast to Western blot, cells transfected with the furin-T2A saRNA had higher RVFV Gn expression compared to those that were transfected with the wild-type consensus saRNA (Figure 5B).

### 3.5. Immunogenicity of the saRNA RVF Candidate Vaccines

Six-week-old BALB/c mice were immunized intramuscularly on day 0 and day 28 with either 0.1 µg, 1.0 µg, or 10 µg of the WT consensus or the furin-T2A saRNA RVF candidate vaccine, and blood samples were collected on days 14, 28, and 42 to measure vaccine-induced humoral immune responses (Figure 6A). RVFV Gn IgG antibodies were detected at all the sample collection time points for all mice immunized with the candidate saRNA RVF vaccines (Figure 6B). For both vaccines, the RVFV Gn IgG antibodies produced were dose-dependent, with their amounts increasing with an increase in the vaccine dose. The two vaccines induced similar quantities of RVFV Gn IgG antibodies, whose amounts were substantially increased after boosting (Appendix A for IgG ELISA raw data files).

RVFV pseudovirus-neutralizing activity was assessed on day 42, two weeks after the booster immunization. Mice immunized with the 10 µg of the WT consensus vaccine had the highest RVFV pseudovirus-neutralizing activity (IC50 = 551, SD = 522, *p* < 0.01) followed by those that received 1 µg of the same vaccine (IC50 = 507, SD = 371, *p* < 0.01). No RVFV pseudovirus-neutralizing activity was detected in the unvaccinated mice or those immunized with 0.1 µg and 1.0 µg of the furin-T2A vaccine. In contrast, RVFV pseudovirus-neutralizing activity was induced in the mice vaccinated with 0.1 µg of the WT consensus and 10 µg of the Furin-T2A vaccine but this was not statistically significant when compared to the unimmunised mice (*p* < 0.05). (Appendix A for RVFV pseudovirus neutralisation raw data file).

## 4. Discussion

Rift Valley fever is a disease that poses a serious threat to public health and the livestock industry. Unfortunately, there is no licensed RVF vaccine for human use, while approved livestock vaccines are unsuitable for universal adoption. This study designed two LNP-formulated saRNA RVF vaccines based on the VEEV genome and tested their humoral immune responses in BALB/c mice. The first vaccine encoded the consensus RVFV M segment sequence coding for Gn and Gc and the second vaccine encoded this sequence but with cytoplasmic tail modifications designed to increase the cell membrane expression of these glycoproteins as a strategy to enhance the immune response.

This study demonstrated that an LNP-formulated saRNA RVF vaccine encoding Gn and Gc induces high quantities of anti-Gn IgG and significant RVFV pseudovirus-neutralizing activity in mice even at a low dose of 1 µg. Numerous candidate RVF vaccines have been developed using various platforms, including virally delivered alphavirus RNA replicons and LNP-formulated conventional mRNA [25,26,27]. To the best of our knowledge, however, this is the first study to report the development of a non-virally delivered (LNP formulated) saRNA RVF vaccine. In general, saRNA vaccines that utilize the VEEV and SINV vector genomes have been shown to have the most promising vaccine immunogenicity [10]. Both of these alphavirus genomes were shown to efficiently express Gn after one immunization but only the VEEV genome induced immune responses that protected the mice against virulent RVFV ZH501 challenge [26].

While the mechanistic correlates of immune protection against RVFV following vaccination are not fully known, the generation of neutralizing antibodies against Gn and Gc has provided a good correlate of protection [28]. It has been shown that RVF vaccines based on Gn alone induce high titers of neutralizing antibodies [29,30]. However, such vaccines may also induce very low or no RVF-neutralizing antibodies [31,32]. It was also reported that an LNP-formulated mRNA vaccine encoding full-length Gn and Gc induced the strongest humoral immune response in mice compared to constructs that expressed Gn alone, Gc alone, or partial sequences of these glycoproteins [27]. Therefore, a design that includes both Gn and Gc appears to be the most suitable for developing an immunogenic RVF vaccine. It should also be noted that while the furin-T2A and H727A constructs had the highest Gn and Gc expressions, respectively, we selected the former along with the WT consensus for synthesis into saRNA. This is because Gn-specific monoclonal antibodies have been demonstrated to have much higher neutralizing activities in vitro and protection efficacy compared to Gc-specific monoclonal antibodies [33]. The WT consensus saRNA enabled us to test the hypothesis that increasing the cell surface expression of Gn and Gc increases the immunogenicity of the RVFV envelope.

This study also demonstrated that the deletion of the Gn and Gc cytoplasmic tails along with the introduction of a furin enzyme and T2A self-cleaving peptide between them increases the extracellular expression of Gn. Surprisingly, the increased extracellular expression of Gn does not lead to a higher increase in serum IgG concentration and also results in a significant reduction in the RVFV pseudovirus-neutralizing activity. We postulate that mutations that increase the extracellular expression of the viral surface glycoproteins lead to conformational changes that hinder the production of effective RVFV pseudovirus-neutralizing antibodies.

The RVFV WT consensus sequence clustered with sequences from the recent RVF outbreaks in East Africa, Madagascar, and Saudi Arabia. This is because these sequences, particularly those from the 2006–2007 Kenya outbreak, dominate the GenBank and ViPR databases that were used to generate the consensus. The greater number of RVF sequences from these outbreaks is due to the increased sequencing and upload of these sequences to these databases. It also reflects the re-emergence or spread of RVF in these countries at the dawn of the 21st century. Despite variations in RVFV strains, RVF vaccines have been successfully used to control disease outbreaks caused by strains that are different from those used in the vaccine. For example, the Smithburn vaccine was successfully used to control RVF outbreaks in Saudi Arabia, Kenya, and East Africa, while clone 13 was used in South Africa [34]. Studies have also shown that antigenic domains in Gn and Gc are well conserved among RVFV strains and that neutralizing antibodies that react to different Gn and Gc epitopes can neutralize all wild-type RVFV strains [35]. This suggests that an RVF vaccine based on a Gn and Gc consensus sequence is likely to induce antibodies that can neutralize all currently circulating RVFV strains.

The main limitation of this study is that only RVFV Gn IgG-binding antibodies and RVFV pseudovirus-neutralizing activity were used to determine the immunogenicity of our candidate vaccines. Other evaluations such as plaque reduction neutralization tests and virus challenge efficacy studies were not performed due to not having the appropriate biosafety containment facilities to conduct these tests. Considering that the immune correlates of protection in RVFV are not fully known, a broader assessment of the vaccine-induced immune responses that include T cell responses along with viral challenge studies would have provided a more comprehensive assessment of the protective mechanisms of our LNP-formulated saRNA RVF vaccine candidates. Nevertheless, neutralizing antibodies that target Gn and Gc have been demonstrated to prevent viral infection, with their titers correlating with protection against virulent RVFV challenges [14,15,16]. Therefore, this study focused on assessing humoral immune responses induced by our candidate saRNA RVF vaccines.

In conclusion, this study showed that an LNP-formulated saRNA RVF vaccine utilizing the VEEV genome to encode full-length Gn ad Gc is a promising RVF vaccine candidate. Further development of this vaccine by assessing its cellular immunogenicity and efficacy is thus warranted.

## Figures and Tables

**Figure 1 vaccines-12-01088-f001:**
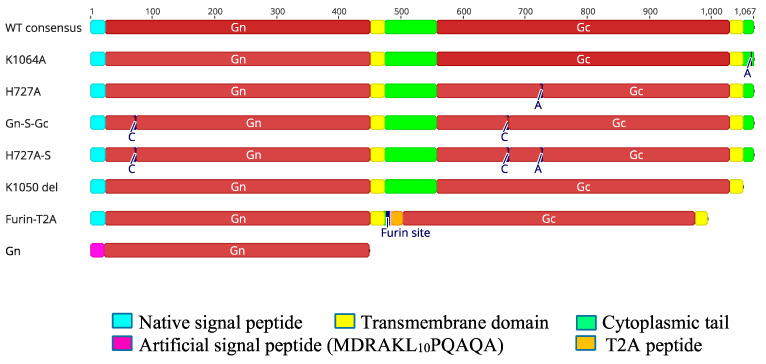
Mutations made to the RVFV M segment consensus sequence.

**Figure 2 vaccines-12-01088-f002:**
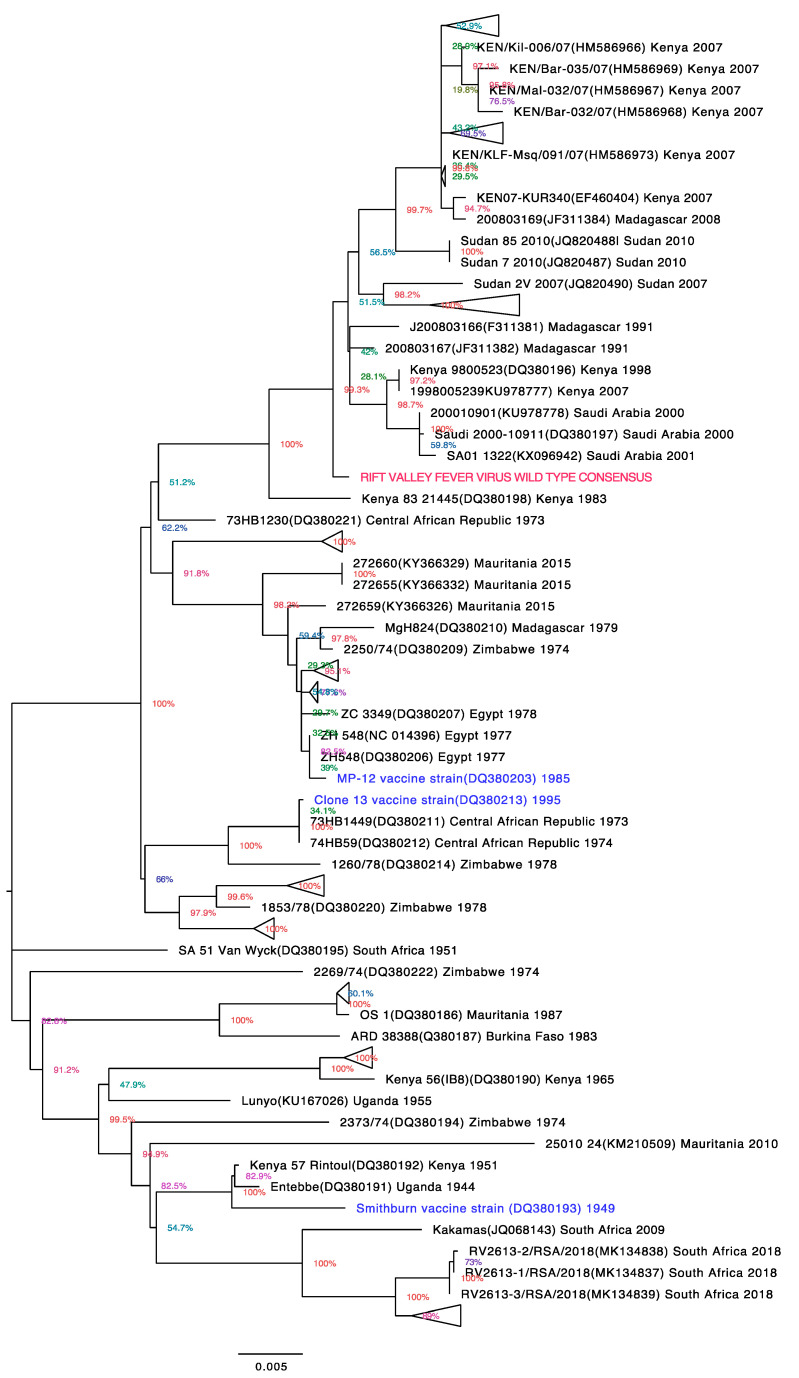
Phylogenetic analysis of the RVFV M segment consensus sequence. The evolutionary history was inferred using the maximum likelihood method and the Tamura 3-parameter model. The tree is drawn to scale, with branch lengths measured in the number of substitutions per site (next to the branches). This analysis involved 158 nucleotide sequences. All positions with less than 95% site coverage were eliminated. There was a total of 3204 positions in the final dataset. Evolutionary analysis was conducted in MEGA X. The tree was edited using FigTree v1.4.4. Tips are labeled with RVFV strain name, GenBank accession number (in brackets), country of origin, and date of isolation. Triangular tips represent similar strains collapsed together to reduce clutter. The tree is unrooted. Sequences highlighted in blue are from RVFV strains of the live attenuated RVF vaccines Smithburn and Clone 13 which have been widely used and MP-12 which has been extensively evaluated.

**Figure 3 vaccines-12-01088-f003:**
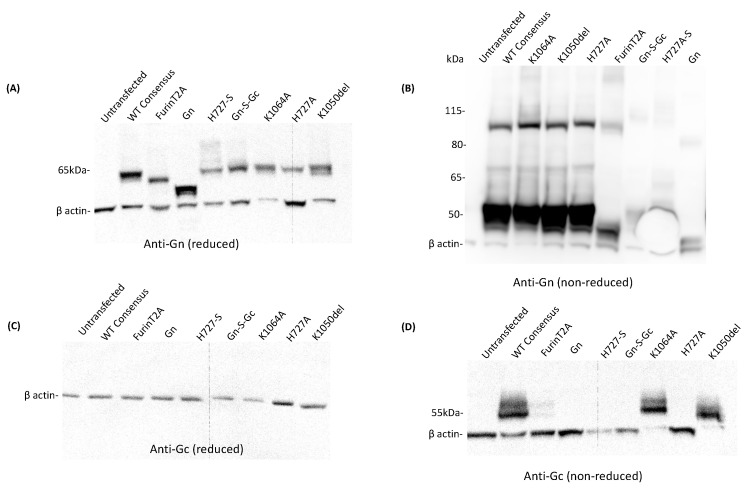
In vitro expression of Gn and Gc from whole-cell lysates using SDS-PAGE Western blot. (**A**) RVFV Gn expression from reduced cell lysates. (**B**) RVFV Gn expression from non-reduced cell lysates. (**C**) RVFV Gc expression from reduced cell lysates (**D**) RVFV Gc expression from non-reduced cell lysates. The samples were reduced by heating at 70 °C in dithiothreitol (DTT) and reoxidation was prevented by using an antioxidant in the electrophoresis and transfer buffers.

**Figure 4 vaccines-12-01088-f004:**
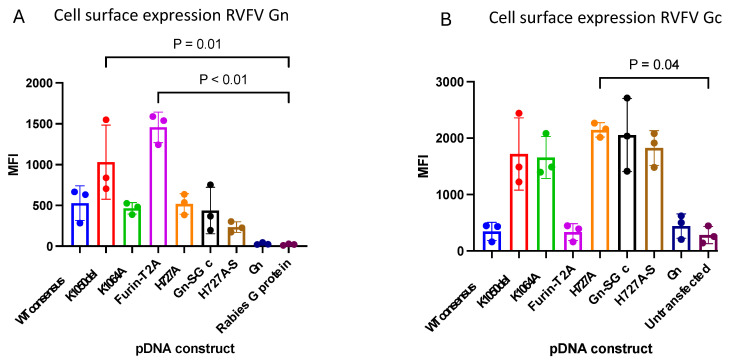
Cell surface expression of RVFV Gn and Gc by flow cytometry. (**A**) Cell surface expression of RVFV Gn. (**B**) Cell surface expression of RVFV Gc. Flow data were analyzed using FlowJo™ v10.10.0 software for Mac (BD Life Sciences) and expression was measured as median fluorescence intensity (MFI). Statistical analysis was performed using GraphPad Prism version 10.1.0. The Kruskal–Wallis test corrected for multiple comparisons using the Dunn’s test was used to compare expression between constructs. A *p*-value of <0.05 was considered statistically significant.

**Figure 5 vaccines-12-01088-f005:**
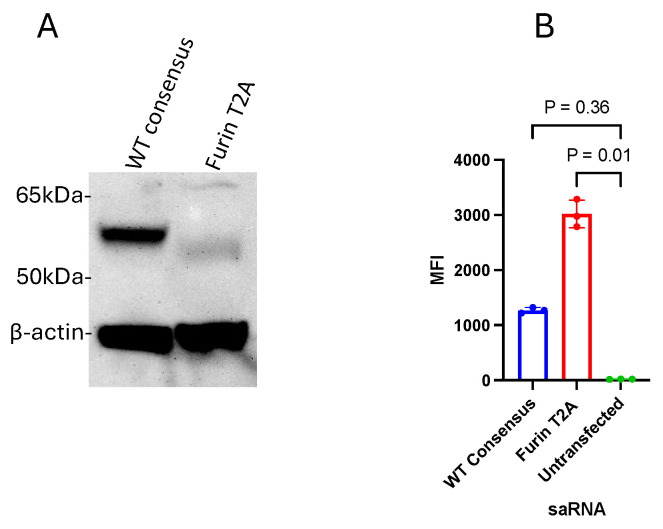
In vitro expression of RVFV Gn from HEK293 cells transfected with RVFV saRNA. (**A**) RVFV Gn expression by SDS-PAGE Western blot of reduced whole-cell lysates. (**B**) Cell surface expression of RVFV Gn by flow cytometry. Flow data were analyzed using FlowJo™ v10.10.0 software for Mac (BD Life Sciences) and expression was measured as median fluorescence intensity (MFI). Statistical analysis was performed using GraphPad Prism version 10.1.0. The Kruskal–Wallis test corrected for multiple comparisons using the Dunn’s test was used to compare expression between constructs. A *p*-value of <0.05 was considered statistically significant.

**Figure 6 vaccines-12-01088-f006:**
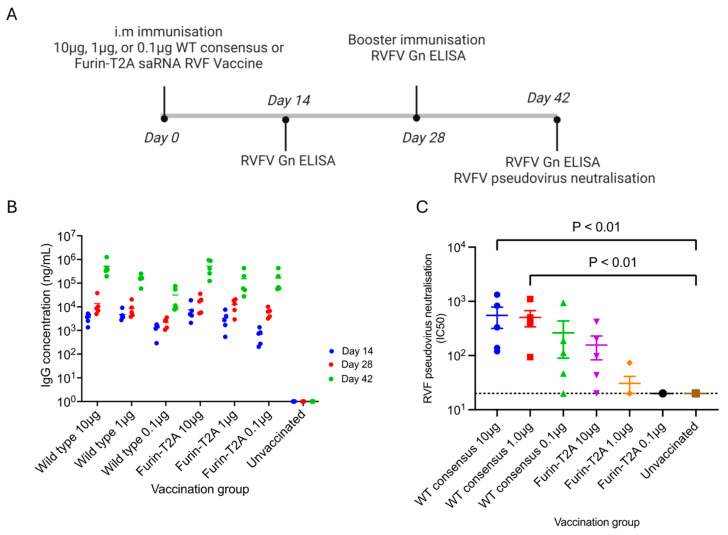
Humoral immune responses in serum of mice immunized with the different saRNA RVF candidate vaccines. (**A**) BALB/c mice were immunized intramuscularly on day 0 and day 28 with 0.1 µg, 1.0 µg, or 10 µg of the WT consensus or the furin-T2A saRNA RVF candidate vaccine. (**B**) serum anti-RVFV Gn IgG concentration on days 14, 28, and 42 for the different vaccination groups. (**C**) RVFV pseudovirus-neutralizing activity on day 42 for the different vaccination groups measured as half-maximal inhibitory concentration (IC_50_).

## Data Availability

The original contributions presented in the study are included in the article/Appendix A; further inquiries can be directed to the corresponding author/s.

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
