# Peer review of "A Lipid Nanoparticle-Formulated Self-Amplifying RNA Rift Valley Fever Vaccine Induces a Robust Humoral Immune Response in Mice"

_vaccines, 2024, doi:10.3390/vaccines12101088_

Round 1
Reviewer 1 Report
Comments and Suggestions for Authors
The manuscript by Kitandwe et al reported a lipid nanoparticle (LNP) formulated self-amplifying RNA vaccine candidate for Rift Valley fever (RVFs). To date, there is no licensed RVF vaccine although some related candidates are in laboratory and clinical trials. These authors provided data that the RVF viral replicon RNA delivered by lipid nanoparticles is able to induce obvious humoral response in mice as well as an obvious RVFV pseudovirus neutralising activity. However, due to biosafety constrains, no viral challenge experiments are conducted, thus the protective effect cannot be concluded. Alternatively, the authors challenged animal with pseudovirus and reported an obvious neutralizing activity. Despite the efforts made in vitro and in vivo, the current form of the manuscript does not meet the quality of Vaccines. Some major concerns towards the data presentation and methodologies are as follows:
1. The saRNA design and selection rationale is not clear. It lacks a detailed rationale for choosing the WT consensus and Furin-T2A designs. In figure 3 and 4, where the expression of Gn and Gc by pDNA in cell lysates as well as cell surface respectively, the WT consensus does not generate more proteins than the others. While in figure 5, only WT and Furin-T2A in vitro expression data by saRNA delivery is shown. What happens to the other constructs? Considering that the surface expression or secreted protein would be more potent antigen for eliciting the humoral immune responses, one would expect the data from other construct for a better comparison. The authors need to provide clear data that support the selection of both candidates over the others.
2. For both WT and Furin-T2A data, it also lacks certain explanation. As shown in figure 5B where both constructs generate similar lgG, while the neutralizing activity differs significantly even same dosage of vaccine is administrated (e.g. WT-1.0ug vs Furin-2T 1.0ug figure 5C). The Gn construct of both design is the same thus it seems that the neutralizing activity is not statistically related to the concentration of IgG as well as the expression efficacy of the saRNA. The authors need to better interpret the data and infer the possible reason. Extra experiments included other constructs in determining the relation between IgG concentration and neutralizing activity would be of great interest.
3. The manuscript focuses on the humoral immune response induced by the RVF replicon vaccine. However, a comprehensive evaluation of the cellular immune response, including T cell activation (like CD4+ and CD8+), cytokine production (e.g., IFN-γ, TNF-α, IL-2), as well as cytotoxic potential, is crucial for understanding the full immunogenic profile of the vaccine candidate.
4. It lacks viral challenge data in animals to assess the protective effect. The authors also recognized this limit and discussed that it due to the lacking of proper biosafety research facilities. But without such experiments, the current story in my opinion is not complete, and does not fit Vaccines here.
The current manuscript should be carefully revised to address the above concerns before consideration for publishing in Vaccines.
With the above major reasons, I would recommend that the current form of the manuscript should be carefully revised before consideration for publication in Vaccines.
Author Response
Thank you very much for taking the time to review this manuscript. Please find the detailed responses below and the corresponding revisions/corrections highlighted/in track changes in the revised manuscript.
Comment 1
The saRNA design and selection rationale is not clear. It lacks a detailed rationale for choosing the WT consensus and Furin-T2A designs. In figure 3 and 4, where the expression of Gn and Gc by pDNA in cell lysates as well as cell surface respectively, the WT consensus does not generate more proteins than the others. While in figure 5, only WT and Furin-T2A in vitro expression data by saRNA delivery is shown. What happens to the other constructs? Considering that the surface expression or secreted protein would be more potent antigen for eliciting the humoral immune responses, one would expect the data from other construct for a better comparison. The authors need to provide clear data that support the selection of both candidates over the others.
Response 1
This study tested the hypothesis that increasing cell surface expression of Gn and Gc increases the immunogenicity of the RVFV envelope. The wild type consensus was therefore preselected for synthesis into saRNA along with one other construct that had the highest in vitro cell surface expression of either Gn or Gc. Even though the construct H727A had the highest Gc expression, we selected the construct with the highest Gn expression FurinT2A for saRNA synthesis. This because Gn-specific monoclonal antibodies have been demonstrated to have much higher neutralizing activity and protective efficacy compared to the Gc-specific monoclonal antibodies. We have clarified this in the results under In vitro expression of RVFV Gn from saRNA and in the third paragraph of the discussion.
Comment 2
For both WT and Furin-T2A data, it also lacks certain explanation. As shown in figure 5B where both constructs generate similar lgG, while the neutralizing activity differs significantly even same dosage of vaccine is administrated (e.g. WT-1.0ug vs Furin-2T 1.0ug figure 5C). The Gn construct of both design is the same thus it seems that the neutralizing activity is not statistically related to the concentration of IgG as well as the expression efficacy of the saRNA. The authors need to better interpret the data and infer the possible reason. Extra experiments included other constructs in determining the relation between IgG concentration and neutralizing activity would be of great interest.
Response 2
Even though the WT consensus and the furin-T2A constructs have the same Gn sequence, we postulate that mutations in the latter led to conformational changes that hindered the production of effective RVFV pseudovirus-neutralizing antibodies. This has been added to the discussion in paragraph four.
Comment 3
The manuscript focuses on the humoral immune response induced by the RVF replicon vaccine. However, a comprehensive evaluation of the cellular immune response, including T cell activation (like CD4+ and CD8+), cytokine production (e.g., IFN-γ, TNF-α, IL-2), as well as cytotoxic potential, is crucial for understanding the full immunogenic profile of the vaccine candidate.
Response 3
Wheras the correlates of immune protection against RVFV infection are not fully understood, neutralising antibodies that target Gn and Gc have been demonstrated to prevent viral infection with their titers correlating with protection against virulent RVFV challenge. Therefore, the evaluation of immune responses induced by our saRNA RVF vaccine candidates primarily focused on their ability to generate humoral immunity. This has been highlighted in the study limitation in the second last paragraph of the discussion.
Comment 4
It lacks viral challenge data in animals to assess the protective effect. The authors also recognized this limit and discussed that it due to the lacking of proper biosafety research facilities. But without such experiments, the current story in my opinion is not complete, and does not fit Vaccines here.
Response 4
The lack of challenge studies is indeed a major limitation for our work and this was highlighted in the discussion. As noted in response to comment 3, neutralising antibodies targeting Gn and Gc epitopes have been demonstrated to prevent RVFV infection. We therefore believe that the induction of RVFV neutralizing activity by our candidate saRNA RVF vaccines is a good indicator of their potential efficacy even in the absence of challenge studies. This has been highlighted in the study limitation in the second last paragraph of the discussion.
Reviewer 2 Report
Comments and Suggestions for Authors
The manuscript by Kitandwe et al. introduces a novel approach to developing Rift Valley Fever (RVF) vaccines using engineered lipid nanoparticles that incorporate various mutated versions of the M segment from the RVF virus, fused with an alphavirus genome containing self-amplifying RNA replicons. This innovative strategy aims to enhance the immune response against RVF, a significant threat to human health. While the work is foundational and offers a promising direction for vaccine development, the manuscript could benefit from more detailed explanations of the experimental designs to aid understanding.
Concerns:
Purpose of Mutations in Figure 1: The authors should explain the rationale behind the design of these specific mutations in the M segment. It would be beneficial to discuss how these mutation sites were selected and what the expected impact of each mutation on the viral protein function or structure might be.
Phylogenetic Analysis Conclusions: The authors should clearly state the conclusions drawn from the phylogenetic analysis.
Reduced Environment in Figure 3: The manuscript should clarify what is meant by a "reduced" and how it impacts the expression of the Gn and Gc proteins. Specific details about whether the reduced conditions were applied during cell lysis or protein loading would help clarify the experimental setup.
Protein Size Differences in Figure 3A: An explanation is needed for why the FurinT2A and Gn constructs show different sizes for Gn expression.
Use of Rabies G Protein in Figure 4A: The rationale for using the Rabies G protein in these experiments should be addressed. If the protein serves as a control or a comparative benchmark, the authors should justify its inclusion and discuss its relevance to the RVF vaccine development.
Significance Testing in Figure 5B: The authors need to provide statistical evidence to support the claim of no significant difference between the WT consensus and the untransfected controls. If the visual data suggest otherwise, this discrepancy must be addressed either by reevaluating the statistical analysis or by providing a clearer graphical representation.
Comments on the Quality of English LanguageThe English is OK, but needs more explanations and descriptions for result interpretation and experimental design.
Author Response
Thank you very much for taking the time to review this manuscript. Please find the detailed responses below and the corresponding revisions/corrections highlighted/in track changes in the revised manuscript.
Comment 1
Purpose of Mutations in Figure 1: The authors should explain the rationale behind the design of these specific mutations in the M segment. It would be beneficial to discuss how these mutation sites were selected and what the expected impact of each mutation on the viral protein function or structure might be.
Response 1
The legend in figure one explains the rationale of each of these mutations. As pointed out in the methods section (Generation of mutated RVFV M segment sequences), the purpose of these mutations was to increase the plasma membrane expression of Gn and Gc or to alter their conformation as a strategy to increase the immunogenicity of these glycoproteins.
Comment 2
Phylogenetic Analysis Conclusions: The authors should clearly state the conclusions drawn from the phylogenetic analysis.
Response 2
These results suggest that the wild-type consensus vaccine is more representative of circulating RVFV strains from recent outbreaks, compared to the older strains used to develop current live attenuated RVF vaccines, such as those from the 1940s (Smithburn) and 1970s (Clone 13 and MP12). Therefore, a vaccine based on the wild-type consensus sequence is more likely to induce antibodies that will most effectively neutralize RVFV strains most relevant to public health and the livestock industry today. This has been added to the manuscript.
Comment 3
Reduced Environment in Figure 3: The manuscript should clarify what is meant by a "reduced" and how it impacts the expression of the Gn and Gc proteins. Specific details about whether the reduced conditions were applied during cell lysis or protein loading would help clarify the experimental setup.
Response 3
Reduced means that the samples were heated at 700C for 10 min with dithiothreitol (DTT) to break the disulfide bonds. Sample reoxidation was prevented by using an antioxidant in the electrophoresis and transfer buffers. We have revised the manuscript and added these details in the legend of Figure 3 and in the methods in the subsection Assessment of in vitro expression using SDS-PAGE and western blot.
Comment 4
Protein Size Differences in Figure 3A: An explanation is needed for why the FurinT2A and Gn constructs show different sizes for Gn expression.
Response 4
An explanation for the size differences between FurinT2A and Gn has been added in the results section under the subsection In vitro expression of RVFV pDNA using SDS-PAGE and western blot. As shown in Figure 1, the Gn only construct lacked both the cytoplasmic tail and transmembrane region while furin-T2A lacked most of its cytoplasmic tail. Therefore, the size of the Gn expressed from these two constructs is expected to be smaller than that of the other constructs.
Comment 5
Use of Rabies G Protein in Figure 4A: The rationale for using the Rabies G protein in these experiments should be addressed. If the protein serves as a control or a comparative benchmark, the authors should justify its inclusion and discuss its relevance to the RVF vaccine development.
Response 5
The rabies G protein plasmid was simply used as a negative control for the experiment and is not of any special relevance to RVF vaccine development. This plasmid has been used before by this research group when developing saRNA vaccines for other pathogens such as for SARS-COV-2 (McKay, P.F., Hu, K., Blakney, A.K. et al. Self-amplifying RNA SARS-CoV-2 lipid nanoparticle vaccine candidate induces high neutralizing antibody titers in mice. Nat Commun 11, 3523 (2020). https://doi.org/10.1038/s41467-020-17409-9)
Comment 6
Significance Testing in Figure 5B: The authors need to provide statistical evidence to support the claim of no significant difference between the WT consensus and the untransfected controls. If the visual data suggest otherwise, this discrepancy must be addressed either by reevaluating the statistical analysis or by providing a clearer graphical representation.
Response 6
This data has been re-analyzed using the Mann-Whitney test, the Kruskal-Wallis test with Dunn’s multiple comparisons, and the one-way ANOVA test, all performed with GraphPad Prism software for Mac, version 10.3. The Mann-Whitney test showed no statistically significant difference between either the wild type or the furin T2A and the control group. The Kruskal-Wallis test showed a statistically significant difference between the furin-T2A and control group while the One-way ANOVA showed a statistically significant difference between both the wild type and furin-T2A and the control group. For this analysis, the Kruskal-Wallis test was used because it is the most suitable one. The Mann-Whitney is most appropriate for comparing only two groups while the One-way Annova is not suitable for analysing small sample sizes. The graph has been revised to show actual P values obtained from the Kruskal-Wallis test.
Reviewer 3 Report
Comments and Suggestions for Authors
The study investigates the immunogenicity of a self-amplifying RNA (saRNA) vaccine for Rift Valley fever (RVF) formulated in lipid nanoparticles. The vaccine encodes the RVF virus (RVFV) medium segment consensus sequence and its derivatives, aiming to enhance cell membrane expression of viral glycoproteins. The saRNA vaccines induced high levels of RVFV Gn IgG antibodies and significant pseudovirus neutralizing activity in mice, demonstrating robust humoral immune responses. The findings suggest that the WT consensus saRNA RVF vaccine is a promising candidate for further development.
Minor comments:
- saRNA were verified by transfection using messenger kit. How about LNP-formulated saRNA? Are they transfected into cells after formulation and how robust it is compared to messenger RNA kit.
- Challenge studies, cellular immunity are all limitations. But the advantages of saRNA are not fully conveyed in my mind. Dosage, long-term immunity, etc. and the disadvantages as well (size limitation, immunogenicity of replicase proteins, etc.
Author Response
Thank you very much for taking the time to review this manuscript. Please find the detailed responses below and the corresponding revisions/corrections highlighted/in track changes in the revised manuscript.
Comment 1
saRNA were verified by transfection using messenger kit. How about LNP-formulated saRNA? Are they transfected into cells after formulation and how robust it is compared to messenger RNA kit.
Response 1
In vitro transfection of cells was only done using the Lipofectamine MessengerMAX kit and not with LNPs. Transfection is done on unformulated saRNA. Lipofectamine MessengerMAX generally gives higher in vitro transfection efficiency compared to LNPs although the former is also more toxic. Lipofectamine MessengerMAX is therefore used only for in vitro transfection while LNPs are used for vivo delivery but may also be used for in vitro transfection
Comment 2
Challenge studies, cellular immunity are all limitations. But the advantages of saRNA are not fully conveyed in my mind. Dosage, long-term immunity, etc. and the disadvantages as well (size limitation, immunogenicity of replicase proteins, etc.
Response 2
We have revised paragraph four of the introduction to highlight the advantages of saRNA
Round 2
Reviewer 1 Report
Comments and Suggestions for Authors
The revised version of the manuscript has made responses to the previous comments accordingly. Especially to comments 1 and 2, where the authors have provided more details to explain the methodology and interpret the data. The authors also emphasized the limitation of the current study in the revised version. I am satisfied with the authors responses and have no further questions. Text and language editing is needed before considering for publishing.
Reviewer 2 Report
Comments and Suggestions for Authors
Thanks for authors' responses which help to better understand the conclusions of this work.